# Exploring the Effects of *VgPIP1;2* Overexpression in the Roots of Young Rice Plants: Modifications in Root Architecture, Transcriptomic and Metabolomic Profiles

**DOI:** 10.3390/plants14233628

**Published:** 2025-11-28

**Authors:** Ítalo Vinícius Cantanhêde Santos, Paloma Koprovski Menguer, Bruno Silvestre Lira, Natalia Balbinott, Felipe Klein Ricachenevsky, Danilo de Menezes Daloso, Magdalena Rossi, Marcia Margis-Pinheiro, Rogério Margis, Helenice Mercier

**Affiliations:** 1Departamento de Botânica, Instituto de Biociências, Universidade de São Paulo, São Paulo 05508-090, SP, Brazil; italosantos@usp.br (Í.V.C.S.); bslbsl@usp.br (B.S.L.); mmrossi@usp.br (M.R.); 2Departamento de Genética, Instituto de Biociências, Universidade Federal do Rio Grande do Sul, Porto Alegre 91501-970, RS, Brazil; paloma.menguer@ufrgs.br (P.K.M.); marcia.margis@ufrgs.br (M.M.-P.); rogerio.margis@ufrgs.br (R.M.); 3Centro de Biotecnologia, Universidade Federal do Rio Grande do Sul, Porto Alegre 91501-970, RS, Brazil; felipecruzalta@gmail.com; 4Departamento de Botânica, Instituto de Biociências, Centro de Biotecnologia, Universidade Federal do Rio Grande do Sul, Porto Alegre 91501-970, RS, Brazil; 5LabPlant, Department of Biochemistry and Molecular Biology, Federal University of Ceará, Fortaleza 60440-900, CE, Brazil; daloso@ufc.br; 6Departamento de Biofísica, Universidade Federal do Rio Grande do Sul, Porto Alegre 91501-970, RS, Brazil

**Keywords:** aquaporin, bromeliad, rice, metabolomics, nitrogen metabolism, root development

## Abstract

Rice (*Oryza sativa*) is a major staple crop that feeds over half of the world’s population. However, its cultivation depends heavily on nitrogen fertilizers, which increase both environmental impacts and production costs. Enhancing the sustainable use of nitrogen is therefore essential for maintaining global food security. Previously, we characterized an aquaporin (VgPIP1;2) from the bromeliad *Vriesea gigantea* that transports ammonium and has great biotechnological potential. Here, we investigated the effect of VgPIP1;2 heterologous expression on rice, particularly in root development and nitrogen metabolism. Transgenic plants cultivated in hydroponics exhibited a larger root network area compared to wild type plants. Biochemical and metabolomic analyses revealed that the roots of VgPIP1;2 overexpressing plants have higher contents of nitrogen, free amino acids and sugars. In line with these results, the transcriptional profile showed that genes involved with nitrogen uptake and assimilation, amino acid biosynthesis and sugar metabolism are upregulated in transgenic plants. These findings indicate that VgPIP1;2 overexpression positively modulates nitrogen and carbon metabolism, altering root development in rice. Thus, the expression of VgPIP1;2 would represent a potential strategy to develop new rice cultivars with improved root architecture suited to enhance nitrogen absorption and assimilation.

## 1. Introduction

Rice (*Oryza sativa* L.) is one of the most important crop species for the human diet, considered a staple food for nearly half of the human population, providing 19% of the caloric energy supply consumed by humans worldwide [1]. It is the third most cultivated plant in the world, with an estimated annual production of over 770 million tons, grown on approximately 160 million hectares of land, resulting in an annual gross production value of over USD 300 billion [2].

Rice cultivation is predominantly conducted in lowland flooded areas, where the soil remains submerged until harvest [3]. This type of management extracts large amounts of nutrients from the soil, which can significantly affect nutrient cycling and crop productivity [4]. Among the supplemented nutrients, nitrogen (N) is one of the most important, not only due to its direct relation to plant productivity [5] but also because rice requires a high nitrogen supply during crop cycle, approximately 1 kg of N for every 40 kg of grain [6].

The uptake of N by rice plants, together with nutrient losses from flooded soils, results in a substantial reduction in N availability within cultivated environments [7]. One of the strategies used by farmers to maximize N absorption by rice plants is the application of high amounts of fertilizers to the soil, in concentrations above those required for rice growth [6,8]. As a result, it is estimated that rice cultivation requires 15% of all N-based fertilizers applied in agriculture worldwide, making it one of the main costs of grain production [9]. In addition, the excessive use of N fertilizers in rice cultivation can lead to various environmental problems, such as greenhouse gas emissions, eutrophication of surface waters, algal blooms, changes in the structure of ecological communities, loss of biodiversity, etc. [10,11].

To exacerbate this situation, the scenario of climate change reduces N availability, resulting in significant impacts on grain productivity [12]. Therefore, the search for strategies to increase N absorption from soils is extremely important for reducing the environmental and economic impacts caused by the use of N fertilizers during rice cultivation, especially in lowland flooded areas [13,14]. In the past few decades, there have been substantial research efforts to develop sustainable agriculture forms, less dependent on N fertilizers [15,16,17,18].

Roots play a central role in determining the efficiency of nutrient acquisition, acting as the primary interface between plants and the soil environment [19]. The architecture and physiology of roots directly influence the plant’s capacity to absorb and assimilate N, especially under fluctuating nutrient conditions typical of flooded soils [15]. Thus, in crops highly dependent on N, such as rice, the improvement of root systems aiming to enhance N uptake represents a key strategy toward sustainable agriculture [20,21].

The genetic manipulation of N metabolism at the absorption, assimilation and/or translocation level, has yielded promising results in terms of increasing N availability to rice plants [22,23,24,25,26,27,28,29]. However, most reports have only focused on the manipulation of rice endogenous N metabolic genes. The few studies that used genes from other species also focused on the expression of N metabolic genes. Accordingly, the biotechnological potential embedded in the functional biodiversity of non-crop species has remained almost entirely unexplored.

The Bromeliaceae family, in particular, displays remarkable physiological diversity in N absorption mechanisms, ranging from the direct uptake of inorganic N forms from the soil by the roots of terrestrial species to the assimilation of atmospherically derived N in epiphytic species [30]. An example of the latter is *Vriesea gigantea* Gaudichaud, an epiphytic bromeliad native to the Brazilian Atlantic Forest [31]. In addition to its morphoanatomical specializations, such as shoots composed of overlapping leaves that form a tank structure, and the specialized trichome on the leaves [32,33,34], previous research from our group has shown that this species contains AQUAPORINs (AQPs) that facilitate N transport [35]. One of those, the PLASMA MEMBRANE INTRINSIC PROTEIN 1;2 (VgPIP1;2), is involved in NH_4_^+^ transport across the plasma membrane, indicating that foliar AQPs of *V. gigantea* can maximize N uptake from N-deficient canopies [35].

In this context, we hypothesized that VgPIP1;2 overexpression in rice plants may enhance N absorption and/or assimilation, thereby enhancing N metabolism and other correlated pathways. Thus, we generated rice plants overexpressing VgPIP1;2 and conducted morphological, physiological, and gene expression analyses of their roots, with a particular focus on N-related processes. The data revealed that VgPIP1;2 expression resulted in changes in root architecture, increase in nitrogen and amino acids content, as well as alterations in transcript and metabolite profiles related to nitrogen and carbon metabolisms in the roots of rice young plants grown under hydroponics conditions.

## 2. Materials and Methods

### 2.1. Transgenic Plant Generation

#### 2.1.1. Vector Construction

The full-length coding sequence of VgPIP1;2 (GenBank MH824553.1) was amplified from *V. gigantea* leaf cDNA using the specific primers listed in Appendix A and the following conditions: 25 ng cDNA, 0.1 mM dNTPs, 0.2 mM of each primer, 1 mM MgCl_2_, and 2 U Taq DNA polymerase; the program was as follows: 94 °C for 5 min, 35 cycles of 94 °C for 30 s, 55 °C for 30 s, 72 °C for 60 s, and 72 °C for 10 min. The amplification product was purified with the GFX™ PCR DNA and Gel Band Purification Kit (Sigma-Aldrich^®^, Saint Louis, MO, USA) and cloned into the entry vector pCR™8/GW/TOPO™ (Thermo Fisher Scientific Inc., Waltham, MA, USA). Then, the coding sequence flanked by the att sites was amplified from the entry vector using an M13 primer pair using the same PCR conditions described above. The purified product was recombined into pANIC6A [36] via LR clonase II Plus (Thermo Fisher Scientific Inc., USA) following the manufacturer’s recommendations.

#### 2.1.2. Plant Transformation

*Agrobacterium tumefaciens*-mediated rice (*Oryza sativa* L. ssp. *japonica* cv. Nipponbare) transformation was performed via co-cultivation of embryogenic calli with the AGL1 strain transformed with the construct of interest. Selection and regeneration of the resistant calli was carried out as previously described by Upadhyaya et al. [37]. Three transgenic lines (OE-1, OE-2 and OE-3) constitutively overexpressing the VgPIP1;2 gene under the control of the maize (*Zea mays*) *UBIQUITIN 1 (ZmUbi1)* promoter were obtained. Homozygous T2-generation plants were used for further phenotypic characterization.

#### 2.1.3. Transgene Expression Analysis

The expression level of VgPIP1;2 was verified via quantitative reverse transcription PCR (RT-qPCR). Total RNA was isolated from plant leaves using Trizol™ reagent (Invitrogen™, Waltham, MA, USA), with three biological replicates per genotype, and the first-strand cDNA synthesis was performed with reverse transcriptase enzyme (M-MLV, Invitrogen™, USA). The RT-qPCR reactions were conducted using an Applied Biosystems StepOne Plus device (Thermo Fisher Scientific Inc., USA). All results were expressed via relative quantification to the *OsUBQ* constitutive gene using the 2-ΔΔCt methodology described by Livak and Schmittgen [38]. The primers used for RT-qPCR are listed in Appendix A.

### 2.2. Morphology Analysis

#### 2.2.1. Disinfestation and Germination

For hydroponic cultivation assays, the three transgenic lines and wild type (WT) seeds were disinfested in a laminar flow chamber by immersion in 70% (*v*/*v*) ethanol for 1 min, followed by 2.5% (*v*/*v*) commercial sodium hypochlorite solution for 30 min, then rinsed five times with autoclaved distilled water. For each genotype, ten Petri dishes containing 100 seeds each were prepared. The seeds were kept submerged in water in the dark for 24 h at 25 °C. Afterward, they were placed in Petri dishes containing filter paper moistened with autoclaved distilled water and kept in a growth chamber under a 16 h/8 h photoperiod at 25 °C, with a light intensity of ~100 µmol m^−2^ s^−1^ for 7 days.

#### 2.2.2. Growth Conditions and Sampling

The previously germinated seedlings were hydroponically cultivated for root phenotypic characterization. Seedlings were transferred to a floating mesh system consisting of rectangular Styrofoam bases covered at the bottom with a polyester-based fabric net, which were placed on top of polypropylene trays (40 × 30 × 7 cm). The trays were filled with 4 L of modified Yoshida [39] nutrient solution, based on Sharma et al. [40] and Feng et al. [41], with the following composition: 1.44 mM (NH_4_)_2_SO_4_, 0.35 mM NaH_2_PO_4_.2H_2_O, 0.51 mM K_2_SO_4_, 1 mM CaCl_2_, 1.64 mM MgSO_4_.7H_2_O, 15.02 µM MnCl_2_.4H_2_O, 0.39 µM Na_2_MoO_4_.2H_2_O, 18.89 µM H_3_BO_3_, 0.15 µM ZnSO_4_.7H_2_O, 0.16 µM CuSO_4_.5H_2_O, 20 µM NaFe-EDTA (all from Labsynth, São Paulo, Brazil) and 7.0 µM dicyandiamide (99% purity, Sigma-Aldrich^®^, USA). The pH of the solution was adjusted to 5.0 with HCl. The nutrient solution was renewed every 3 days. Seedlings were maintained under hydroponic conditions in a growth chamber with a 12 h light (~250 µmol m^−2^ s^−1^)/12 h dark photoperiod, at 28 °C during the day and 22 °C at night, with 80% relative humidity during the day and 60% at night, for 10 days.

After this period, the seedlings were transferred to 400 mL polypropylene pots with lids. The plants were secured using polyurethane foam in pre-drilled 3 cm diameter holes in the lids, which were then attached to the pots containing the nutrient solution described above. After 15 days, the roots were scanned using an HP Officejet Pro 8600 scanner, and images were analyzed with RhizoVision Explorer (v.2.0.2) to obtain morphological parameters. The results were obtained from three independent experiments with at least ten plants per genotype.

### 2.3. Biochemical and Gene Expression Analyses

Based on the morphological characterization and transgene expression level, the transgenic line OE-3 was chosen for further biochemical and gene expression analyses. For this, seed germination and hydroponic plant cultivation were carried out as previously described, using Yoshida’s [39] modified nutrient solution with 1.44 mM NH_4_NO_3_ (Labsynth, Brazil) as the N source. After 15 days, the roots were harvested, immediately frozen and stored at −80 °C.

Ammonium sulfate was used in the first experiment to supply a single, defined ammonium source, allowing the assessment of VgPIP1;2 effects under strictly ammonium nutrition. In the second experiment, ammonium nitrate was employed to provide a mixed N source, reflecting the natural coexistence of ammonium and nitrate in paddy soils due to nitrification processes [42]. This adjustment aimed to simulate field-like conditions and to evaluate plant responses under a more physiologically relevant N environment, considering the synergistic effects between both N forms on rice metabolism [43].

#### 2.3.1. Nitrogen Content

The N content in the roots of WT and OE-3 plants was measured using the Pregl-Dumas method with a thermal conductivity detector (PerkinElmer 2400 Series II model) (PerkinElmer, Waltham, MA, USA) based on Moț et al. [44]. The results were obtained from two independent experiments with five plants per genotype.

#### 2.3.2. Ammonium Content

Endogenous ammonium content in roots was measured based on the method described by Vega-Mas et al. [45], with modifications. A 40 mg aliquot of plant material was placed in 1.5 mL microtubes, to which 1 mL of Milli-Q water and one 5 mm tungsten carbide bead was added. The samples were then disrupted in a TissueLyser II system (QIAGEN^®^, Hilden, Germany) for 1 min at a frequency of 30 Hz. Afterward, the tubes were centrifuged at 4000 rpm at 4 °C for 20 min in a Sorvall ST 16R centrifuge (Thermo Fisher Scientific Inc., USA), and the supernatant was collected for the quantification of total ammonium content. In each well of a 96-well plate, 50 μL of plant extract, 100 μL of 0.33 M sodium phenolate (98% purity, Sigma-Aldrich^®^, USA), 50 μL of 0.02% (*v*/*v*) sodium nitroprusside (99% purity, Sigma-Aldrich^®^, USA), and 100 μL of 2% (*v*/*v*) commercial sodium hypochlorite were added. Absorbance was measured at 635 nm using a spectrophotometer plate reader (Epoch—Agilent BioTek, Santa Clara, CA, USA), and ammonium concentrations were calculated based on an (NH_4_)_2_SO_4_ standard curve (ranging from 0.05 to 1 mM). The results were obtained from two independent experiments with five plants per genotype.

#### 2.3.3. Free Amino Acid Content

The total free amino acid content in the roots was quantified according to López-Hidalgo et al. [46]. Fresh plant material (60 mg) and 1 mL of 80% ethanol (*v*/*v*) at 4 °C were added to a 2 mL microtube with one 5 mm tungsten carbide bead added. The samples were then disrupted in a TissueLyser II system (QIAGEN^®^, Germany) for 1 min at a frequency of 30 Hz. Afterward, they were placed in an ice-cooled ultrasonic bath (Thornton Eletrônica Ltd., Vinhedo, Brazil) for 10 min and then centrifuged at 10,000× *g* at 4 °C for 10 min. The supernatant was collected for the quantification of total free amino acids. For this, 150 μL of plant extract was transferred to 1.5 mL safelock microcentrifuge tubes, followed by the addition of 75 μL of 2% ninhydrin reagent solution (Sigma-Aldrich^®^, USA), and mixed with a vortex. The tubes were incubated at 100 °C in a thermal block (AccuBlock™Digital Dry bath D1200, Labnet, Edison, NJ, USA) for 10 min and then cooled on ice for another 10 min. After cooling, 150 μL of 95% (*v*/*v*) ethanol was added to each tube. In triplicate, 150 μL of the reaction mixture was transferred to each well of a 96-well microplate, and absorbance was measured at 570 nm in a plate reader (Epoch—Agilent BioTek, USA). Endogenous amino acid content was calculated using an L-proline (99% purity, Labsynth, Brazil) standard curve (ranging from 0.03 to 1 mg/mL). The results were obtained from two independent experiments with five plants per genotype.

#### 2.3.4. Total Soluble Protein Content

For quantification of total soluble proteins in plant roots, 200 μL of Milli-Q water was added to 1.5 mL microtubes containing 20 mg of plant material. The samples (five per genotype) were then vortex-homogenized for 1 min and centrifuged at 25,000× *g* at 25 °C for 5 min. The supernatant was collected and used for protein quantification based on the Bradford [47] method. Known concentrations of bovine serum albumin (99% purity, Sigma-Aldrich^®^, USA) were used as a standard curve for calculating protein concentrations (10 to 200 μg/mL). The results were obtained from two independent experiments.

#### 2.3.5. Metabolomics Analysis

The extraction of polar metabolites and the gas chromatography mass spectrometry (GC-MS)-based metabolomics analysis were carried out exactly as described previously by Lisec et al. [48]. Approximately 20 mg of lyophilized ground roots were used for the extraction, and each sample comprised two pooled roots derived from distinct independent experiments. Briefly, polar metabolites were extracted using methanol (LC-MS grade, Sigma-Aldrich^®^, USA), water and chloroform (LC-MS grade, Sigma-Aldrich^®^, USA). A total of 150 µL of the polar phase was dried in a SpeedVac and derivatized using methoxyamine hydrochloride (dissolved in 20 mg/mL in pure pyridine) and N-methyl-N-(trimethylsilyl)trifluoroacetamide (MSTFA) (98,5% purity, Sigma-Aldrich^®^, USA). Derivatized samples were injected in a GC-MS (QP-PLUS 2010, Shimadzu, Kyoto, Japan). The mass spectral analysis was carried out using Xcalibur 2.1 software (Thermo Fisher Scientific), and the metabolites were identified using the Golm Metabolome Database (http://gmd.mpimp-golm.mpg.de/ (accessed on 18 March 2025) [49].

#### 2.3.6. Transcriptomics Analysis

Total RNA was purified from four biological replicates per genotype (transgenic line OE-3 or WT), each being composed of roots from four plants from the same experiment, using a Direct-zol^™^ RNA MiniPrep Plus kit (Zymo Research^®^, Irvine, CA, USA). RNA-seq libraries were prepared and sequenced using the TruSeq Stranded mRNA Library Prep kit at Macrogen Inc. (Seoul, Republic of Korea) and the Illumina NovaSeq platform (Illumina Inc., San Diego, CA, USA).

#### 2.3.7. Sequence Analyses, Differential Expression and Gene Ontology Enrichment

The quality of RNA-seq raw reads was assessed with FastQC software, and Trim Galore! was used to remove adapter sequences and trim low-quality reads, retaining reads with quality scores above Q30 and a minimum length of 85 nucleotides for further analyses. The retained high-quality reads were then mapped against the *Oryza sativa* MSU7.0 genome assembly using HISAT2 [50]. Differential expression analysis was performed using SARtools with the DESeq2 algorithm [51,52], and *p*-values were adjusted using Benjamini–Hochberg’s correction [53]. Genes with *p*-values < 0.05 were considered differentially expressed, regardless of fold change (FC). Genes were categorized as up- or down-regulated based on their expression changes across the different genotypes. Gene Ontology (GO) and Kyoto Encyclopedia of Genes and Genomes (KEGG) pathway enrichment analyses were conducted on the differentially expressed genes using ShinyGO 0.77 [54]. This analysis used the genes identified in all libraries as a reference background, applying a false discovery rate (FDR) cutoff of 0.05 to identify significantly enriched terms. The graphs were generated using the online platforms iDEP: integrated Differential Expression and Pathway analysis (version 2.0) [55], ShinyGO (version 0.77) and SRplot [56].

### 2.4. Phylogenetic Analysis

The phylogeny of VgPIP1;2 was inferred using the maximum likelihood method and Jones–Taylor–Thornton [57] model of amino acid substitutions, and the tree with the highest log likelihood (−15,888.25) is shown. The percentage of replicate trees in which the associated taxa clustered together (200 replicates) is shown next to the branches [58]. The initial tree for the heuristic search was selected by choosing the tree with the superior log-likelihood between a Neighbor-Joining (NJ) tree [59] and a Maximum Parsimony (MP) tree. The NJ tree was generated using a matrix of pairwise distances computed using the p-distance [60]. The MP tree had the shortest length among 10 MP tree searches, each performed with a randomly generated starting tree. In plants, aquaporins are grouped into major subfamilies. The PIPs (PLASMA MEMBRANE INTRINSIC PROTEINS), TIPs (TONOPLAST INTRINSIC PROTEINS), and NIPs (NODULIN-LIKE INTRINSIC PROTEINS), each characterized by distinct subcellular localizations and substrate transport specificities. The analytical dataset comprised 82 amino acid sequences obtained from UniProt (https://www.uniprot.org (accessed on 18 October 2025), representing the three aquaporin subfamilies: 18 from *Ananas comosus* (Ac), 32 from *Arabidopsis thaliana* (At), 29 from *Oryza sativa* var. Nipponbare (Os) and 3 from *Vriesea gigantea* (Vg). The partial deletion option was applied to eliminate all positions with less than 95% site coverage, resulting in a final data set comprising 223 positions. Evolutionary analyses were conducted in MEGA12 [61] utilizing up to 6 parallel computing threads.

### 2.5. Statistical Analysis

Statistical analyses of germination, hydroponic cultivation and biochemical parameters were performed using IBM^®^ SPSS^®^ Statistics version 21 and GraphPad Prism version 10.1.2. The experimental design was completely randomized. The data were subjected to the Shapiro–Wilk test (*p* > 0.05) to verify their normality. Statistical differences between groups were determined either by Student’s *t*-test or two-way Analysis of Variance (ANOVA) followed by Tukey’s HSD test (a = 0.05) when the data showed normal distribution, or Mann–Whitney or Kruskal–Wallis tests if the data distribution was not normal. Metabolomics data were further analyzed by Principal Component Analysis (PCA) and Partial Least-Squares Discriminant Analysis (PLS-DA) using the MetaboAnalyst platform [62]. The PLS-DA also provided a list of metabolites with variable importance in projection (VIP) scores, which indicates the importance of each metabolite for the PLS model. Metabolites with VIP score greater than 1 were considered significant for the PLS model, as previously suggested by Xia and Wishart [63]. Thus, metabolites with a VIP score higher than 1 were those that mostly explained the results observed in the PLS-DA.

## 3. Results

### 3.1. VgPIP1;2-Overexpressing Lines and Changes in Root Architecture

Three VgPIP1;2-overexpressing lines, namely OE-1, OE-2 and OE-3, were obtained. While OE-1 and OE-2 displayed similar amounts of VgPIP1;2 mRNA, OE-3 exhibited significantly higher expression of the transgene (Figure 1). As expected, no transgene expression was observed in the WT plants. No upregulation was detected in the expression of *OsPIP1;2* and *OsPIP1;3* in the transgenic plants (Appendix A), which are the closest rice orthologs of VgPIP1;2 (Appendix A).

The roots of the transgenic lines grown under hydroponic conditions exhibited a similar root architecture among themselves, which differed from that of the wild-type plants. They developed longer roots with greater diameter and dry mass, resulting in a larger root network area compared to the wild type plants (Figure 2). Since the three VgPIP1;2-overexpressing lines exhibited a similar root phenotype, OE-3 line was chosen for further biochemical and gene expression analyses.

### 3.2. VgPIP1;2 Overexpression Promotes Changes in the Metabolite Profile

After 25 days of hydroponic cultivation, the roots of transgenic plants accumulated higher N and free amino acid contents. However, no significant differences were found in the amounts of ammonium and total soluble proteins (Figure 3).

Regarding the metabolites identified by GC-MS, there was an overall trend of increases in the contents of the identified metabolites in the OE-3 line in comparison to the WT, including most of the amino acids, in agreement with the spectrometry quantification mentioned above. Among the 43 metabolites identified, the levels of aspartate (Asp), trehalose, gamma-aminobutyric acid (GABA), and pyroglutamate were significantly increased in the transgenic line (Figure 4A). Despite this low number of significantly altered metabolites, both supervised (PLS-DA) and unsupervised (PCA) multivariate analyses clearly separated OE-3 and WT genotypes (Figure 4B and Appendix A), demonstrating that transgenic plants also differ from the WT at the metabolic level. Seventeen metabolites had VIP scores higher than 1. GABA, trehalose, pyroglutamate, Asp, glycine, threonic acid, fructose, and sucrose had VIP scores higher than 1.4 (Figure 4C), thus being the metabolites that mostly explain the separation observed in the PLS-DA graph (Figure 4B).

### 3.3. VgPIP1;2 Overexpression Promotes Changes in Gene Expression Profile

To understand the origin of the altered metabolic profile observed in the OE-3 line described above, we conducted RNA-seq analysis in the same root samples used for GC-MS from the transgenic and WT genotypes. The sequencing yielded a total of 240 million paired-end raw reads (100 bp each). After filtering the data to remove adapter sequences and low-quality reads, approximately 211 million high-quality reads were obtained, representing 87% of the total raw reads. About 188 million reads were aligned to the rice genome (*Oryza sativa* ssp. *japonica* cv. “Nipponbare” v. MSU7), accounting for 89% of the total sequences obtained after data filtering. Of the 42,189 protein-coding genes in the reference genome, 30,219 were expressed in at least one of the sequenced samples.

Principal component analysis (PCA) showed that the genotypes differed from each other in terms of global gene expression profile (Figure 5A). We identified 3059 differentially expressed genes (DEG), with 1244 downregulated and 1815 upregulated in the OE-3 line compared to the WT (Figure 5B).

To identify putative alterations in pathways or biological processes, the previously identified DEGs were subjected to enrichment analyses based on GO and KEGG terms (Figure 5C–F). Among the enriched terms, we noticed that most of them were primarily related to N, carbon (C), and lipid metabolism. The most enriched GO term based on the upregulated genes was “Nitrate metabolic process”. Among the enriched metabolic pathways for KEGG, terms related to N metabolism were also identified, such as “Nitrogen metabolism”, the third most enriched term. Additionally, pathways related to amino acid metabolism were enriched in the analysis, including “Beta-Alanine metabolism”, “Pantothenate and CoA biosynthesis” and “Biosynthesis of amino acids”.

Several pathways related to C metabolism were also enriched. The “Carbohydrate metabolic process” was the most enriched term directly associated with C metabolism for GO. A series of KEGG terms, such as “Glycerolipid metabolism”, “Galactose metabolism”, “Amino sugar and nucleotide sugar metabolism”, “Starch and sucrose metabolism” and “Carbon metabolism” were also enriched based on the upregulated genes in the transgenic plants. The second most enriched GO term was “Fatty acid metabolic process”. Six metabolic pathways related to lipid metabolism were also enriched from the list of upregulated genes in KEGG. The most prominent of these was “Fatty acid elongation”, followed by “Biosynthesis of unsaturated fatty acids”.

These data suggest that, as hypothesized, VgPIP1;2 overexpression increases the levels of N in the roots of transgenic plants, reprograming several metabolic pathways.

### 3.4. VgPIP1;2 Overexpression Causes Changes in Nitrogen and Carbon Metabolism

Based on the metabolome and transcriptome results, we decided to take a closer look at N metabolism, specifically regarding the absorption and assimilation of N, and synthesis of amino acids such as Asp and GABA, as well as C metabolism, with an emphasis on the trehalose synthetic pathway (Figure 6 and Appendix A).

We observed a noticeable increase in the expression of genes encoding NITRATE TRANSPORTER (*OsNRT2.1*, *OsNRT2.2*, *OsNRT2.3*, and *OsNRT2.4*) and AMMONIUM TRANSPORTER (*OsAMT1.3* and *OsAMT3.3*) in OE-3 roots. Genes encoding various enzymes related to ammonium assimilation were also differentially regulated between the genotypes. Two genes encoding the cytosolic and plastidial forms of the GLUTAMINE SYNTHETASE (GS) enzyme (*OsGS1;1* and *OsGS2*) and the gene encoding the GLUTAMATE SYNTHASE (GOGAT) dependent of ferredoxin (*OsFd-GOGAT*) were upregulated in the roots of OE-3 plants. The GLUTAMATE DECARBOXYLASE (*OsGAD1*) enzyme-encoding gene, responsible for the synthesis of GABA from glutamate (Glu), was upregulated as well (Figure 6A). No differences were observed in the expression profile of genes encoding ASPARTATE AMINOTRANSFERASE isoforms, despite the marked difference in Asp levels between the genotypes.

We also observed differences in the gene expression profile and metabolite abundance related to sugar metabolism. Overall, there was an upward tendency in the levels of saccharides such as glucose, fructose, sucrose, and particularly trehalose. There were also some differences in the expression of genes related to the trehalose biosynthesis pathway. At least one INVERTASE (INV), SUCROSE SYNTHASE (SUS) and TREHALOSE-6P SYNTHASE (TPS) encoding genes were upregulated. Intriguingly, two genes encoding TREHALOSE-6P PHOSPHATASE (*OsTPP4* and *OsTPP6*) were strongly repressed in the transgenic plants, contrasting with the elevated level of this disaccharide in this genotype (Figure 6B).

## 4. Discussion

In recent years, multiple studies have focused on improving the use of N in rice [22,24,27,29,64,65]. Here, a novel approach has been adopted, exploring an adaptative strategy of the epiphytic bromeliad *Vriesea gigantea*, a Brazilian native species. The VgPIP1;2 gene, which encodes an aquaporin that facilitates both water and N transport, was overexpressed in rice.

Previous studies conducted by our research group have shown the various adaptations that epiphytic bromeliads possess, which allow them to cope with the low nutrient availability in the epiphytic environment [66,67,68,69,70]. In *V. gigantea*, one of these adaptations is related to the function of the VgPIP1;2 aquaporin identified as a facilitator of N transport in the form of ammonium into the cells [35].

The heterologous expression of VgPIP1;2 enabled the growth of colonies of a *Saccharomyces cerevisiae* strain defective in ammonium transport under low concentrations, highlighting the role of this aquaporin in N transport. Phylogenetic analyses showed that VgPIP1;2 was grouped into the PIP1 subclass and exhibited a high degree of identity with PIP1;2 from *Ananas comosus* (94.5%) [35]. The aquaporin gene VgPIP1;2 from *Vriesea gigantea* clusters within the PIP1 subgroup together with *Oryza sativa* and *Arabidopsis thaliana* orthologs (Appendix A). The topology supports the evolutionary conservation of PIP1-type aquaporins among monocot and dicot species, reinforcing the structural and functional similarity between VgPIP1;2 and its rice homolog *OsPIP1;2*.

The overexpression of *OsPIP1;2* has already been used with the aim of improving rice growth; however, the transgenic plants exhibited higher mesophyll conductance of CO_2_ and increased CO_2_ assimilation with no impact on N content and metabolism [71]. Similar results were obtained for *PIP1;2* from *Arabidopsis thaliana* and *Nicotiana tabacum* [72,73].

Although VgPIP1;2 encodes an aquaporin that facilitates ammonium transport in *V. gigantea* [35], no differences were observed in the endogenous ammonium levels in the transgenic plants compared to WT counterparts. Due to the toxic nature of ammonium accumulation [74,75,76], it is expected that, once absorbed by roots, NH_4_^+^ is rapidly assimilated into amino acids via the GS/GOGAT cycle, preventing its harmful effects in high concentrations. Indeed, higher amino acid amounts were verified in the roots of VgPIP1;2 expressing plants, with no impact on total protein content, suggesting that this increase results from enhanced de novo biosynthesis rather than from protein degradation.

VgPIP1;2 expression in rice plants led to significant modifications in root system architecture, including increased length, diameter, and total root area compared to wild-type plants. These results are likely a consequence of enhanced nitrogen uptake and signaling caused by VgPIP1;2 with improved acquisition of ammonium and nitrate. Since ammonium stimulates lateral root branching and nitrate promotes elongation, the simultaneous enhancement of both pathways likely produced a synergistic effect on root development [77].

This combined effect of N forms on root architecture is usually mediated by auxins [42]. Indeed, an upregulation of genes involved in the auxin response pathway was observed in the roots of transgenic plants (see Appendix A), indicating the participation of this hormone in the modulation of root morphology. Consequently, the transgenic plants exhibited a more extensive root system, reflecting a coordinated adjustment of root architecture to improved nitrogen availability. Similar root morphological improvements have been observed in rice plants overexpressing N transport- or metabolism-related genes [17,27,65].

The root transcriptome analysis revealed several findings that help to better understand the phenotypes observed in transgenic plants. Among the N-metabolism-related genes upregulated in the roots of the transgenic plants, *OsGS1;1* and *OsGS2*, as well as *OsFd-GOGAT*, stood out. The expression of these genes is closely linked to ammonium availability and absorption, as they encode enzymes responsible for ammonium assimilation into amino acids [78,79,80]. The positive regulation of these genes in the transgenic plants reinforces the idea that plants overexpressing VgPIP1;2 have enhanced ammonium absorption.

In addition to NH_4_^+^ assimilation genes, those involved in nitrate transport and assimilation were also upregulated in the roots of transgenic plants, indicating an increase in nitrate uptake. A synergistic effect between the absorption/assimilation of ammonium and nitrate has been reported [77]. These two molecules have opposing effects on intracellular pH, as NH_4_^+^ absorption can cause alkalinization of the cytosol, while nitrate absorption can lead to its acidification [81,82].

Changes in cellular pH can be sensed by cells and lead to alterations in nitrate assimilation [83,84]. It has been demonstrated that the OsNRT2.3 transporter can sense intracellular pH changes through a pH-sensitive motif located in the cytosolic portion of the protein [85]. Therefore, we speculate that enhanced ammonium uptake induced by VgPIP1;2 increased the intracellular pH, what was then detected by the pH-sensitive motif of OsNRT2.3, leading to the upregulation of nitrate transporters, thereby facilitating greater nitrate influx into root cells and balancing cellular charges. In this sense, several genes involved in redox regulation were also modulated in the transgenic roots (see Appendix A).

The second most abundant amino acid in the transgenic plants was Asp. This amino acid is a key metabolic regulator due to its role in the crossroad between C and N metabolisms [86]. Higher Asp concentrations in plants have been associated with increased N use efficiency and greater plant productivity [87]. Another metabolite identified as upregulated in the metabolome analysis is GABA, a non-protein amino acid formed from the decarboxylation of Glu by the enzyme glutamate decarboxylase [88]. GABA synthesis can serve as a mechanism for assimilating excess production of Glu through its decarboxylation [89,90]. The increased concentration of GABA in the roots of transgenic plants indicates a potential escape route from Glu synthesis via the GS/GOGAT cycle.

C metabolism was also modulated in transgenic plants. We observed a trend toward an increase in the levels of sucrose, glucose and fructose in the roots of transgenic plants overexpressing VgPIP1;2. N availability can positively influence sucrose transport through the phloem [26,91]. In rice, the expression of sucrose transporter genes—such as SUCROSE TRANSPORTER (*OsSUT*) and SUGARS WILL EVENTUALLY BE EXPORTED TRANSPORTERS (*OsSWEET*), mediated by transcription factors such as DNA BINDING WITH ONEFINGER (OsDOF)—has been associated with increased amino acid synthesis and root growth [91,92,93]. In fact, one isoform of *SWEET* (*OsSWEET2)* was upregulated in OE-3 plants (see Appendix A). Thus, the increase in amino acid levels could have enhanced sugar transport via the phloem from shoot to roots, supplying C scaffolds to sustain amino acid synthesis and also influencing in the root architecture of transgenic plants.

The C balance between source and sink organs may be related to trehalose levels and the expression of genes related to its synthesis. Trehalose is a disaccharide synthesized from UDP-glucose and glucose-6-phosphate through the actions of two enzymes, TREHALOSE-6-PHOSPHATE SYNTHASE (TPS) and TREHALOSE-6P PHOSPHATASE (TPP) [94]. Increasing evidence highlights the crucial role of the trehalose precursor, trehalose-6-P, as a signaling molecule responsible for maintaining C balance in plants. Generally, higher levels of trehalose-6-P promote an increase in sucrose synthesis and transport to sink organs, while lower levels tend to have the opposite effect [95,96].

Particularly, in relation to root development, changes in morphology happen in combination with changes in auxin transport [97]. In *Arabidopsis*, reduced expression of the *AtTPPI* gene stunted primary root length and compromised auxin transport through a reduction in the expression of auxin transporters (PIN 1 and PIN 3) [98]. Nutrient supply can also have an impact on trehalose metabolism. In *Arabidopsis*, Blasing et al. [99] reported that nitrate repressed *AtTPPC* and induced *AtTPPB* and *AtTPPE*, suggesting an interaction between trehalose and N metabolism. This N effect could be important for the control of C/N homeostasis *in planta* [97].

Unfortunately, trehalose-6-P was not detectable in metabolomic analyses due to its low concentration in plant tissues. However, the expression of trehalose metabolic genes from the RNA-seq analysis can help us to understand the signaling role of this metabolite in the transgenic rice roots. The upregulation of a TPS isoform gene was observed alongside strong downregulation of TPP genes. Moreover, we observed an increase in trehalose levels in transgenic plants. These results suggest that trehalose-6-P levels may have been modulated in OE-3 roots, potentially correlating with an increase in sucrose transport from shoot to roots.

## 5. Conclusions

Based on the results obtained in this study, we conclude that the overexpression of VgPIP1;2 modulates root morphology as well as the transcriptomic and metabolomic profiles of rice. N metabolism was strongly affected, with increased N and free amino acid contents in the roots, accompanied by the upregulation of genes involved in N uptake and assimilation. Together, these findings support our initial hypothesis that introducing VgPIP1;2 into rice plants enhances N acquisition and assimilation, ultimately promoting beneficial changes in root development that may improve N use efficiency in the soil.

## Figures and Tables

**Figure 1 plants-14-03628-f001:**
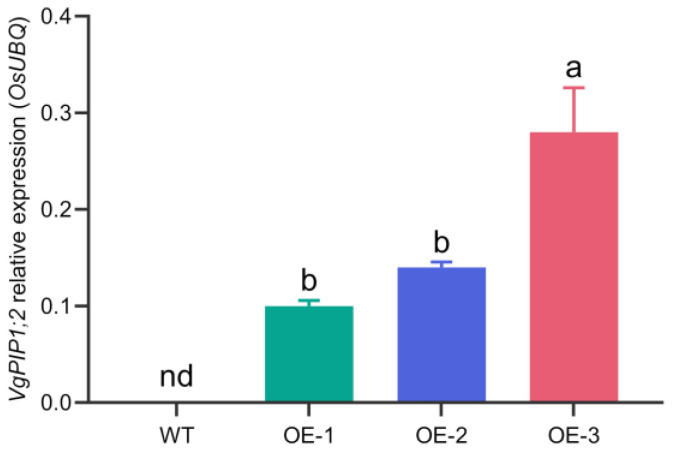
Transgene expression level in rice lines constitutively overexpressing VgPIP1;2. Relative VgPIP1;2 expression compared to *OsUBQ* constitutive gene. Values represent mean ± SEM (*n* = 3). Different letters indicate significant differences among genotypes (*p* < 0.05).

**Figure 2 plants-14-03628-f002:**
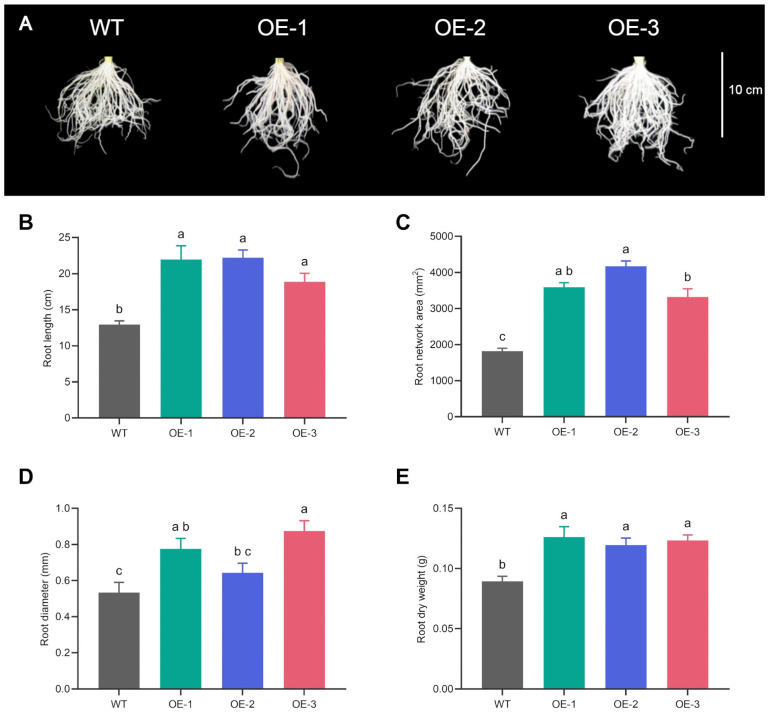
Overexpression of VgPIP1;2 promotes changes in the root architecture of rice plants. (**A**) Root phenotypes. (**B**) Root length. (**C**) Root network area. (**D**) Root diameter. (**E**) Root dry weight. Values represent mean ± SEM (*n* = 10). Different letters indicate significant differences among genotypes (*p* < 0.05).

**Figure 3 plants-14-03628-f003:**
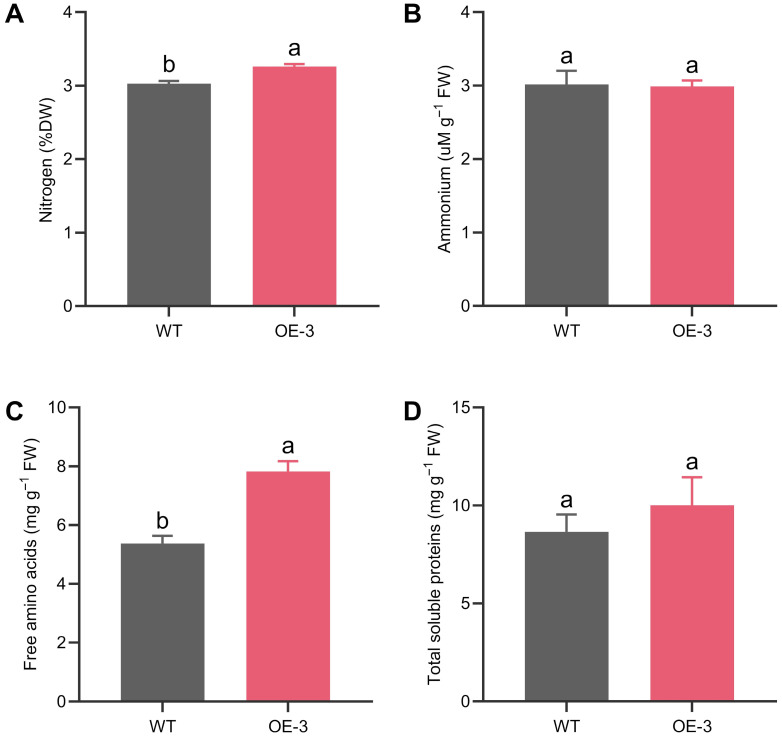
Overexpression of VgPIP1;2 increases nitrogen and amino acid levels in the roots of transgenic rice plants. Quantification of (**A**) nitrogen, (**B**) ammonium, (**C**) total free amino acids and (**D**) total soluble proteins in the roots of hydroponic-cultivated OE-3 line and WT plants. Values represent mean ± SEM (*n* = 5). Different letters indicate significant differences among genotypes (*p* < 0.05).

**Figure 4 plants-14-03628-f004:**
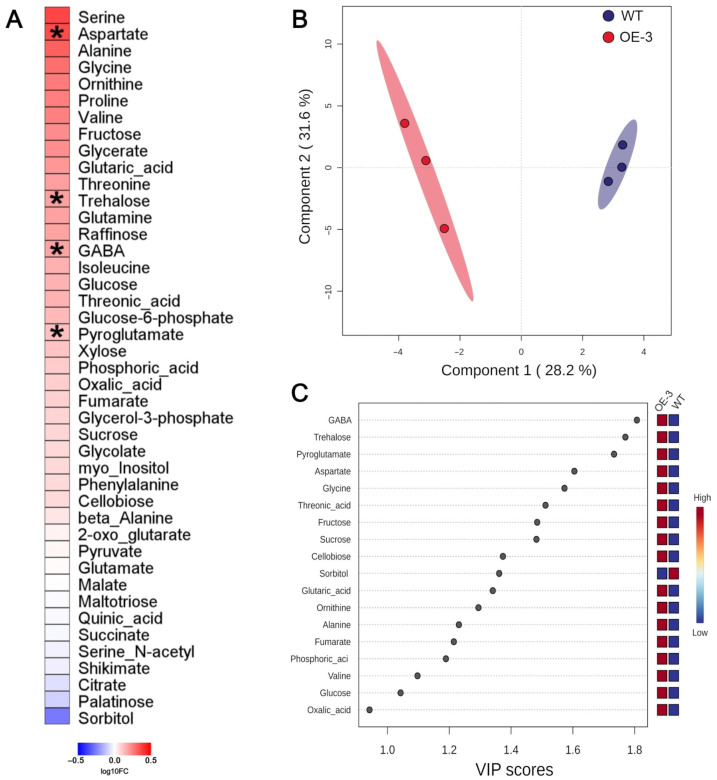
Overexpression of VgPIP1;2 alters the global profile of polar metabolites in rice roots. (**A**) Heat map represents the relative abundance of the identified metabolites in OE-3 line compared to WT genotype. Asterisks denote significant differences among genotypes (*p* < 0.05). (**B**) Partial least squares-discriminant analysis (PLS-DA) of metabolites identified in root samples of OE-3 line and WT plants. (**C**) Variable importance in projection (VIP) scores from the PLS-DA model.

**Figure 5 plants-14-03628-f005:**
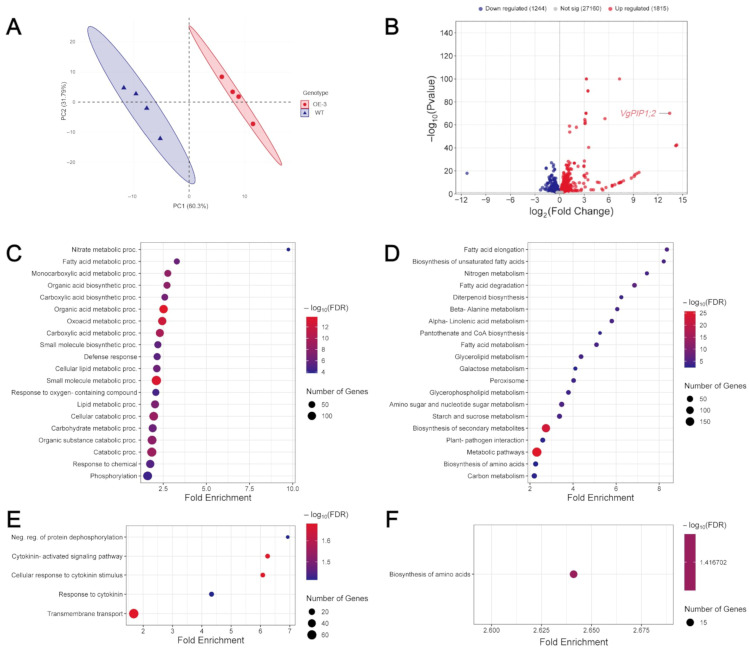
Overexpression of VgPIP1;2 alters the global gene expression profile in rice roots. (**A**) Principal component analysis (PCA) of the root transcriptional profiles of OE-3 line and WT plants. (**B**) Volcano plot showing the numbers of downregulated and upregulated differentially expressed genes (DEGs) in the OE-3 line compared to WT plants. Functional enrichment analysis of DEGs based on Gene Ontology (GO) biological process (**C**,**E**) and Kyoto Encyclopedia of Genes and Genomes (KEGG) molecular pathways (**D**,**F**). Enrichment analysis for upregulated (**C**,**D**) or downregulated (**E**,**F**) genes. Only the top 20 false discovery rate (FDR) ranked terms are shown (FDR cutoff = 0.05).

**Figure 6 plants-14-03628-f006:**
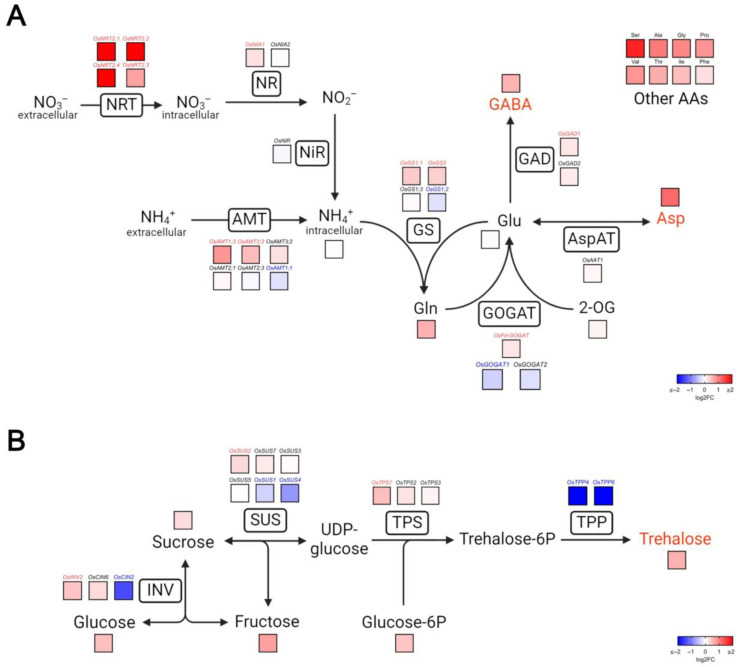
Overexpression of VgPIP1;2 induces changes in gene expression and abundance of metabolites related to N and C metabolism in rice roots. Schematic representation of nitrogen (**A**) and carbon metabolism (**B**), highlighting nitrate and ammonium uptake, amino acid synthesis and trehalose biosynthetic pathways. Heat maps represent the relative abundance of genes or metabolites in OE-3 line compared to WT genotype. Names written in red, blue or black indicate genes and metabolites significantly upregulated, downregulated or invariable, respectively, in OE-3 plants (*p* < 0.05). Names written in rectangles represent enzymes or transporters, while names outside rectangles represent metabolites within each pathway. NRT: NITRATE TRANSPORTER, AMT: AMMONIUM TRANSPORTER, NR: NITRATE REDUCTASE, NIR: NITRITE REDUCTASE, GS: GLUTAMINE SYNTHETASE, GOGAT: GLUTAMATE SYNTHASE, ASPAT: ASPARTATE AMINOTRANSFERASE, GAD: GLUTAMATE DECARBOXYLASE, INV: INVERTASE, SUS: SUCROSE SYNTHASE, TPS: TREHALOSE 6-PHOSPHATE SYNTHASE, TPP: TREHALOSE 6-PHOSPHATE PHOSPHATASE, Gln: glutamine, Glu: glutamate, Asp: aspartate, GABA: gamma-aminobutyric acid, Ser: serine, Ala: alanine, Gly: glycine, Pro: proline, Val: valine, Thr: threonine, Ile: isoleucine, Phe: phenylalanine, 2-OG: 2-oxoglutarate.

## Data Availability

The original contributions presented in this study are included in the article/Appendix A. Further inquiries can be directed to the corresponding author.

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
