# Peer review of "Exploring the Effects of VgPIP1;2 Overexpression in the Roots of Young Rice Plants: Modifications in Root Architecture, Transcriptomic and Metabolomic Profiles"

_plants, 2025, doi:10.3390/plants14233628_

Round 1

Reviewer 1 Report

Comments and Suggestions for Authors

1 This manuscript displayed the effect of overexpressing a heterologous VgPIP1;2 gene on root development and nitrogen metabolism in young rice roots. The roots of VgPIP1;2 overexpressing plants have higher content of nitrogen, free amino acid and sugars. Genes involved with nitrogen uptake and assimilation, amino acid biosynthesis, and sugar metabolism are upregulated in the transgenic plants. These findings indicate that VgPIP1;2 overexpression positively modulates nitrogen and carbon metabolism, altering root development in rice. This demonstrates the possibility of utilizing this gene to enhance nitrogen uptake in rice genetic improvement. After minor revisions, this manuscript is worthy of publication in this journal.

2 In section “2. Materials and Methods”,please indicate the number of samples / repetitions in each experiment / investigation.

3 in section 2.3.4, please indicate which tissue(s) were(was) used for investigation.

4 In section 2.4, please indicate the data source used for analysis.

5 pleas check the placement of Supplementary Figure 1, and Supplementary Figure 2.  

6 in lines 289-291, please reorganize the sentence “No upregulation was observed in the expression of OsPIP1;2 and OsPIP1;3 in transgenic plants (Supplementary Table 2) the two closest orthologs in the rice genome”, it causes reading difficulties.

Author Response

Thank you very much for taking your time to review this manuscript. We 
appreciate your comments and suggestions; they will be very useful for improving 
the quality of the manuscript.  Please find the detailed responses below and the 
corresponding corrections highlighted in the re-submitted files. 

Comment 1: 2 In section “2. Materials and Methods”, please indicate the 
number of samples / repetitions in each experiment / investigation. 
Response 1: Thank you for pointing this out. We agree with this suggestion and 
we added the required information to the new version of the manuscript. 

Comment 2: In section 2.3.4, please indicate which tissue(s) were(was) used for 
investigation. 
Response 2: Thank you for the observation, the information was added to the 
manuscript. 

Comment 3: In section 2.4, please indicate the data source used for analysis. 
Response 3: The information regarding the data source was added to the text. 

Comment 4: Please check the placement of Supplementary Figure 1, and 
Supplementary Figure 2.   
Response 4: Thank you for pointing out this issue. The order of the 
supplementary figures, as well as their corresponding legends, was changed in 
the new submission of the manuscript. 

Comment 5: In lines 289-291, please reorganize the sentence “No upregulation 
was observed in the expression of OsPIP1;2 and OsPIP1;3 in transgenic plants 
(Supplementary Table 2) the two closest orthologs in the rice genome”, it causes 
reading difficulties. 
Response 5: We appreciate your suggestion. We changed the structure of the 
sentence to make it clearer. 
We once again thank you for your time and for the valuable suggestions provided 
to our manuscript. We hope that the revisions made have improved the text 
making it suitable for publication.

Reviewer 2 Report

Comments and Suggestions for Authors

The manuscript titled “Exploring the effects of VgPIP1;2 overexpression in the roots of young rice plants: modifications in root architecture, transcriptomic and metabolomic profiles” has been reviewed. In the written, some informative results would beneficial us to understand the mechanism of introducing VgPIP1;2 into rice plants may promote differential N uptake and assimilation, leading to positive impacts on rice root development that would result in improved N soil utilization. However, the ultimate result of beneficial effects should be an increase in biomass or individual plant yield, not just root biomass. Therefore, please provide data and images on how genetically modified plants can increase individual biomass or yield.

Author Response

Thank you very much for taking your time to review this manuscript. We 
appreciate your comments and suggestions; they will be very useful for improving 
the quality of the manuscript.  Please find the detailed responses below and the 
corresponding corrections highlighted in the re-submitted files. 
Comment 1: […] However, the ultimate result of beneficial effects should be an 
increase in biomass or individual plant yield, not just root biomass. Therefore, 
please provide data and images on how genetically modified plants can increase 
individual biomass or yield. 
Response 1: Thank you very much for your considerations regarding our 
manuscript. Unfortunately, we do not have these data at the moment, as the 
hydroponic experiment was exclusively designed for the phenotypic, 
transcriptomic, and metabolomic characterization of the roots. However, we are 
currently conducting an independent experiment with adult plants grown in soil 
under greenhouse conditions, in which we have already observed significant 
differences in the shoots of transgenic plants, including increases in the number 
of leaves, tillers, panicles, and in the protein content of the grains (see Figure 1). 
These results, which suggest greater biomass and improved grain quality, will be 
included in an additional manuscript that is currently in preparation. These results 
will be part of another manuscript that is still in preparation. 

We once again thank you for your time and for the valuable suggestions provided 
to our manuscript. We hope that the revisions made have improved the text 
making it suitable for publicati

Reviewer 3 Report

Comments and Suggestions for Authors

Dear Authors!

In this study, the authors investigated the effect of the heterologous aquaporin VgPIP1;2 gene expression on root development and nitrogen metabolism in transgenic rice plants. Biochemical and metabolomic analyses showed that overexpression of VgPIP1;2 gene positively affected nitrogen and carbon metabolism, leading to changes in root development in the rice plants. The findings could be used to help researchers create new rice varieties with improved root systems suitable for enhanced nitrogen uptake and assimilation.

There are some comments for Authors.

- The ‘Materials and methods’ section should be supplemented to allow researchers to repeat the experiments. It is important to describe all reagents used, including the manufacturer (name, country, and degree of purity).

- The country of origin for the equipment used in the experiments should be indicated.

- The article does not specify the brands of PCR thermal cycler, ultrasonic bath, centrifuge (company, country).

- It would be helpful to include the Conclusions chapter in the article.

- In the References chapter, many references contain the names of organisms that should be written in italics.

After considering these comments, this manuscript can be recommended for publication in Plants.

Author Response

Thank you very much for taking your time to review this manuscript. We 
appreciate your comments and suggestions; they will be very useful for improving 
the quality of the manuscript.  Please find the detailed responses below and the 
corresponding corrections highlighted in the re-submitted files. 

Comment 1: The ‘Materials and methods’ section should be supplemented to 
allow researchers to repeat the experiments. It is important to describe all 
reagents used, including the manufacturer (name, country, and degree of purity). 
Response 1: Thank you for pointing this out. We agree with this suggestion and 
we added the required information to the new version of the manuscript. 

Comment 2: The country of origin for the equipment used in the experiments 
should be indicated. 
Response 2: Thank you for the observation, the information was added to the 
manuscript. 

Comment 3: The article does not specify the brands of PCR thermal cycler, 
ultrasonic bath, centrifuge (company, country). 
Response 3: The information regarding the equipment used in the experiments 
was added to the text. 

Comment 4: It would be helpful to include the Conclusions chapter in the article. 
Response 4: We appreciate your suggestion and we added the section 
Conclusions to the manuscript. 

Comment 5: In the References chapter, many references contain the names of 
organisms that should be written in italics 
Response 5: Thank you for pointing out this issue. We corrected the problem in 
the new version of the manuscript. 
We once again thank you for your time and for the valuable suggestions provided 
to our manuscript. We hope that the revisions made have improved the text 
making it suitable for publication.